# Association between Shift Work and Reflux Esophagitis: The Kangbuk Samsung Health Study

**DOI:** 10.3390/ijerph18126189

**Published:** 2021-06-08

**Authors:** Eunchan Mun, Daehoon Kim, Yesung Lee, Woncheol Lee, Soyoung Park

**Affiliations:** 1Department of Occupational and Environmental Medicine, Kangbuk Samsung Hospital, Sungkyunkwan University School of Medicine, Seoul 03181, Korea; e6u3n0@gmail.com (E.M.); daehoon963.kim@samsung.com (D.K.); y313.lee@samsung.com (Y.L.); wonch.lee@samsung.com (W.L.); 2Total Healthcare Center, Kangbuk Samsung Hospital, Sungkyunkwan University School of Medicine, Seoul 04514, Korea

**Keywords:** reflux esophagitis, shift work schedule, circadian rhythm

## Abstract

This cross-sectional study investigated the relationship between shift work and reflux esophagitis verified by endoscopic findings. Participants underwent a comprehensive health examination and esophagogastroduodenoscopy between January 2011 and December 2018. We examined endoscopic findings and performed multivariate-adjusted regression analysis of the association between shift work and reflux esophagitis using multiple demographic and clinical factors. Among the 247,450 participants, 49,767 (20.1%) had reflux esophagitis; Los Angeles (LA)-M (31,132, 12.6%) was most common, followed by LA-A (16,213, 6.6%), LA-B (2333, 0.9%), and ≥LA-C (89, 0.04%). The multivariate-fully adjusted odds ratio (OR) of overall reflux esophagitis for shift work compared to fixed day work was 1.15 (95% confidence interval [CI]: 1.11–1.19). When classified according to shift work type and severity of reflux esophagitis, the ORs of LA-A for regular day and night, and irregular shifts compared to fixed day works were 1.14 (95% CI: 1.03–1.26) and 1.26 (95% CI: 1.11–1.44), respectively. However, there was no significant association between any shift work schedule and ≥LA-B. Overall, we demonstrated the cross-sectional association between shift work (especially rotating and irregular shifts) and mild reflux esophagitis (≤LA-A) compared with daily fixed time shifts.

## 1. Introduction

Traditionally, shift work has been an occupational health issue worldwide, especially among factory workers. However, with the advent of the 24 h society, shift work has extended to those employed in white-collar occupations [1]. Socioeconomic demands for 24 h maintenance services and processes have been increasing, and approximately 20–30% of workers in Europe and the United States are engaged in shift work [2,3]. In addition, approximately 9.7% of employed individuals in South Korea are estimated to be engaged in shift work, and this percentage is increasing steadily [4].

There are extensive publications on the health effects of shift work. Previous studies have shown that shift workers are at increased risk of various health impairments such as coronary heart disease, stroke [5], type 2 diabetes [6], obesity and other metabolic disorders [7], breast cancer [8], colorectal cancer [9], prostate cancer [10], and all-cause mortality [11]. Although the causal pathway has not been fully explored, the possible mechanisms can be summarized as behavioral mechanisms, psychosocial stress, and physiological mechanisms such as disruption of circadian rhythm [12]. Regarding the gastrointestinal system, circadian disruption can decrease circulating melatonin responsible for gut motility, and affect the maintenance of the mucosal barrier, decreasing gastroprotective factors while increasing the secretion of gastric acid [13]. As a result, shift work could increase the number of reflux events, induce damage to the esophagogastric junction, and delay recovery. Nevertheless, studies on gastroesophageal reflux disease in the working environment are lacking.

Gastroesophageal reflux disease (GERD) is a condition that develops when the reflux of stomach contents causes troublesome symptoms and/or complications according to the Montreal definition [14]. Practically, the diagnosis of GERD is usually based on reflux symptoms rather than objective evidence of reflux esophagitis, such as endoscopic findings and 24 h pH monitoring tests, especially in primary care settings [15]. Several studies have shown a link between shift work and reflux symptoms. Xue et al. analyzed a total of 2027 workers who completed a comprehensive health checkup at a university hospital in China, showing that rotating night shift workers had an increased risk of GERD symptoms than day workers (odds ratio [OR]: 3.66; 95% competence interval [CI]: 2.52–5.40) [16]. Li et al. also showed that shift work was an independent risk factor for GERD symptoms (OR: 1.38; 95% CI: 1.11–1.71) in an investigation of 15,283 outpatients at 10 hospitals in China [17]. However, even typical reflux symptoms are not specific for reflux esophagitis, and this may have caused overestimation of the risk of reflux esophagitis [18,19]. Few studies have used objective markers of reflux esophagitis to determine the effect of shift work on reflux esophagitis. This study demonstrated the shift work was positively associated with reflux esophagitis diagnosed using esophagogastroduodenoscopy (EGD).

## 2. Materials and Methods

### 2.1. Study Design and Participants

This cross-sectional study investigated participants in the Kangbuk Samsung Health Study between January 2011 and December 2018 (*n* = 386,488) and followed the Strengthening the Reporting of Observational Studies in Epidemiology (STROBE) guidelines and the STROBE checklist for cross-sectional studies [20]. The Kangbuk Samsung Health Study included a cohort of Korean men and women aged 18 years or older, who underwent comprehensive health screening examinations at the clinics of the Kangbuk Samsung Hospital Total Healthcare Center in Seoul and Suwon, South Korea. More than 80% of participants or their spouses were employees of various companies and local governmental organizations, because the South Korean Industrial Safety and Health Law provides free annual or biennial health screenings for employees. The remaining participants voluntarily purchased the screening examinations. In the case of multiple visits, the first visit was investigated instead of the last visit to prevent any further increase of causal interference in this cross-sectional study. The exclusion criteria were as follows: no endoscopy or endoscopy failure, missing occupational information or not working, a history of cancer derived from the database of the National Cancer Registration and Statistics Program, digestive medication use, and missing covariate data. A total of 247,450 workers were included in the analysis (Figure 1). This study was conducted in accordance with the guidelines of the Declaration of Helsinki of 1975 and was approved by the Institutional Review Board of the Kangbuk Samsung Hospital (approval number: KBSMC 2020-10-020; approval date: 16 October 2020). The requirement for informed consent was waived owing to the use of de-identified data.

### 2.2. Measurement of Types of Shift Work

Data on age, sex, smoking status, alcohol consumption, exercise habits, marital status, education level, abdominal symptoms, medication, and occupational information were obtained using a self-administered questionnaire, and reviewed during a face-to-face interview. Work schedules were investigated with the question, “In the past year, during which time of the day have you worked the most?” The following answers were possible: “I work mostly during the day (between 6 AM and 6 PM)” and “I work during other hours.” Participants who responded to the latter were classified as shift workers and asked to select one of the following six work schedules:Fixed evening shift (between 2 PM and 12 AM);Fixed night shift (between 9 PM and next day AM);Regular day and night shifts;24 h shifts;Split shift;Irregular shift work.

### 2.3. Mesasurement and Classification of Reflux Esophagitis

A standardized endoscopic examination using esophagogastroduodenoscopy was performed as part of a comprehensive health check-up, and the endoscopic findings were recorded and classified by experienced endoscopists. We used the Los Angeles (LA) classification system with Japanese modifications including minimal change grade (LA-M), as noted below (Japanese translated into English) [21]:LA-M: no esophageal mucosal injury, but white turbidity of the mucosa or redness with unclear boundaries that cannot be said to be mucosal damage.LA-A: mucosal injury with a major axis not exceeding 5 mm, limited to the mucosal folds.LA-B: at least one mucosal injury with a major axis of 5 mm or more; mucosal injuries on different mucosal folds are not continuous with each other.LA-C: mucosal injury in at least one place that spreads continuously to two or more folds but involves less than 75% of the circumference.LA-D: continuous mucosal injury involving at least 75% of the esophageal circumference.

### 2.4. Mesasurement of Covariates

Clinical factors such as height, body weight, and waist circumference were measured by trained nurses, and blood parameters were measured using venous blood samples collected from the antecubital vein after ≥10 h of fasting. Serum total cholesterol, high-density lipoprotein cholesterol (HDL-C), low-density lipoprotein cholesterol (LDL-C), and triglyceride levels were measured using an enzymatic colorimetric assay. Serum fasting glucose levels were measured using the hexokinase method on a Cobas Integra 800 apparatus (Roche Diagnostics, Tokyo, Japan). All laboratory tests were conducted at the Laboratory Medicine Department of Kangbuk Samsung Hospital in Seoul, South Korea, which is accredited by the Korean Society of Laboratory Medicine and the Korean Association of Quality Assurance for Clinical Laboratories.

### 2.5. Statistical Analysis

For the descriptions of general characteristics of study participants and the distribution of severity of reflux esophagitis (LA-M, LA-A, LA-B, and ≥LA-C), categorical variables were expressed as a percentage (%) and analyzed using the Chi-squared test. Non-normally distributed continuous variables are expressed as median values with interquartile ranges (IQRs), and were analyzed using the Mann–Whitney U test. A normality test was performed using the Kolmogorov–Smirnov test. Binary logistic regression was used to determine the association between shift work and reflux esophagitis. For subgroup analysis, multinomial logistic regression was used to determine the association between shift work schedules and types of reflux esophagitis (LA-M, LA-A, and ≥LA-B). The results of binary and multinomial logistic regressions are expressed as ORs with 95% CIs. Multiple demographic and clinical screening factors were used for multivariate adjustment, followed by age and sex, as well as personal behavioral factors such as smoking status, alcohol intake, exercise, marital status, and education level for Model 1. Additionally, metabolic factors of glucose, body mass index (BMI), waist circumference, total cholesterol, HDL-C, LDL-C, and triglycerides were further adjusted for Model 2. All statistical analyses were performed using SPSS software (version 24.0; IBM Corp., New York, NY, USA). Statistical significance was set at *p* < 0.05.

## 3. Results

### 3.1. Participant Characteristics

The median age was 36 (IQR, 31–42) years, and the majority (65.0%) of the participants were men. Most were married (74.1%) and well-educated (84.6%) workers. In total, 20.1% of the participants had reflux esophagitis. There were significant differences in age, sex, smoking status, alcohol intake, marital status, education level, BMI, waist circumference, shift work schedules, glucose, total cholesterol, HDL-C, LDL-C, and triglyceride levels between the normal group and reflux esophagitis group. Certain shift work schedules (regular day and night shifts, 24 h shifts, and irregular shift work) were more common in the reflux esophagitis group than in the normal group (Table 1).

### 3.2. The Distribution of Endoscopic Findings of Reflux Esophagitis

Among the 247,450 study participants, 49,767 (20.1%) had reflux esophagitis; LA-M (31,132, 12.6%) was the most common, followed by LA-A (16,213, 6.6%), LA-B (2333, 0.9%), and ≥LA-C (89, 0.04%). All reflux esophagitis types were more common in men. When study participants were stratified by five age groups (18–29, 30–39, 40–49, 50–59, and ≥60 years), LA-M was common in young participants; however, LA-A through ≥ LA-C tended to increase with age (*p* for trends < 0.001) (Table 2).

### 3.3. Association between Shift Work and Reflux Esophagitis

In all regression models, shift work compared to fixed day work was significantly associated with reflux esophagitis (all *p* < 0.001). The magnitude of association was highest in age and sex adjusted model (OR: 1.18, 95% CI: 1.14–1.22) and lowest in the crude model (OR: 1.09 95% CI: 1.05–1.13). Even after full adjustment of all demographic and clinical screening factors including metabolic factors, the association remained consistently significant (OR: 1, 15; 95% CI: 1.11–1.19) (Table 3).

### 3.4. Association between Shift Work Schedules and Types of Reflux Esophagitis

Compared to fixed day work, shift work was significantly associated with the overall types of reflux esophagitis (OR: 1.15; 95% CI: 1.11–1.19), LA-M (OR: 1.17; 95% CI: 1.12–1.21), and LA-A (OR: 1.11; 95% CI: 1.05–1.18) in the full-adjusted model. There was no significant association between shift work and ≥LA-B. For additional classification according to shift work type, regular day and night shifts and irregular shift work were significantly associated with LA-M and LA-A. Twenty-four-hour shifts were only significantly associated with LA-M. However, fixed evening shift, fixed night shift, and split shift were not significantly associated with any type of reflux esophagitis (Table 4).

## 4. Discussion

In this study, we found that shift work was cross-sectionally associated with reflux esophagitis, verified by endoscopy, even after full adjustment for multiple potential confounding factors. In particular, mild degrees of reflux esophagitis were significantly positively associated with rotating or irregular shifts.

It is estimated that the prevalence of GERD is approximately 13% worldwide; however, there are considerable geographic variations. Moreover, there is a possibility of overestimation, as GERD is typically diagnosed based on subjective symptoms without any gold standard for diagnosis [22,23]. Conversely, well-controlled large-scale studies presenting the prevalence of reflux esophagitis diagnosed using objective tests such as endoscopy and 24 h pH monitoring are limited. According to a recent nationwide multicenter study on 2687 Koreans, the prevalence of endoscopic reflux esophagitis of grade ≥LA-A was 9.7% [24]; in our study, the prevalence was 7.5% (*n* = 18,635). The subjects in our study were part of a relatively healthy occupational cohort; this may explain the relatively lower prevalence of erosive esophagitis compared with those of previous studies.

Although the causal pathway of the increased risk of reflux esophagitis among shift workers is not fully understood, several plausible mechanisms have been suggested by previous studies. Currently, circadian disruption caused by shift work is one of the most reliable pathways. Shift work is well known for altering light exposure, diet patterns, and sleep patterns [10], and all of these factors result in circadian rhythm disruption of regulatory hormones, especially melatonin [25]. Melatonin maintains esophageal mucosa integrity through its antioxidative effects against free radical destruction [26] and vasodilating effects on esophageal microcirculation [27]. It also minimizes contact with reflux contents by inhibiting the secretion of acid and pepsin, while stimulating duodenal bicarbonate secretion and increasing gastrin release; this in turn increases contractility of the lower esophageal sphincter (LES) [28,29,30]. Thus, circadian disruption which is responsible for low levels of melatonin, could be a principal factor in increasing the risk of reflux esophagitis among shift workers.

Apart from disrupted circadian rhythms, altered dietary habits as a result of shift work can increase the risk of reflux events. Shift workers tend to snack more and have irregular mealtimes [31,32], and several studies have reported that irregular diet habits, especially short meal-to-sleep intervals, may be related to increased transient LES relaxation [33,34,35]. Additionally, shift workers are likely to adopt unhealthy behaviors, such as reduced physical activity [36], smoking [37], and alcohol use [38]; these may in turn adversely affect reflux esophagitis [39,40,41,42]. Obesity and being metabolically unhealthy are also related to shift work [43], and can increase the risk of reflux esophagitis [44]. However, our study focused on the effects of circadian misalignment, rather than other possible causal pathways. The positive relationship between shift work and reflux esophagitis was consistent even after full adjustment of personal behavior and multiple clinical factors such as BMI, waist circumference, and other metabolic factors.

We stratified the type of shift work and the severity of reflux esophagitis. Rotating or irregular shift work schedules were associated with mild reflux esophagitis, while fixed shift work schedules were not significantly associated with reflux esophagitis. Since fast rotating or irregular shift work may not allow for sufficient time for the intrinsic circadian clock to adapt, more severe disruptions of circadian rhythm may occur and result in lower melatonin levels compared to fixed shift work [45]. In addition, the regularity of dietary habits may have contributed to the effects on reflux esophagitis, which differed by types of shift work. For shift workers, irregular working hours are the most influential factor on the timing of meals. As shift workers who work irregular hours have difficulty in maintaining regular eating patterns, fixed time shifts would have been more beneficial than rotating or irregular shifts for achieving regular dietary habits [46].

In this study, the effect size of shift work for reflux esophagitis was not large because reflux esophagitis can be caused by a number of similarly influential factors rather than very deterministic factors. Therefore, a small effect size of a single factor (shift work) is an expected result [47].

Several previous studies support our results. Chung et al. showed that shift work was cross-sectionally associated with erosive esophagitis in 6040 male shipyard workers (OR, 1.41; 95% CI, 1.03–1.94). However, they did not analyze differences according to the type of shift work schedule or severity of reflux esophagitis. Furthermore, the study participants were all healthy male workers with the same occupational background; generalizability was thus limited [48]. Li et al. verified that shift work is an independent risk factor associated with GERD symptoms through an analysis of 15,283 outpatients (OR: 1.38; 95% CI: 1.11–1.71). However, their study was limited by only analyzing outpatients rather than healthy workers [17]. Xue et al. obtained more generalizable results than previous studies by using health checkup data. They showed that shift work and GERD symptoms are cross-sectionally associated (OR: 3.66, 95% CI: 2.52–5.40); however, their study was limited by the fact that they could not correlate GERD symptoms to endoscopic findings [16].

Our study has some limitations. First, its cross-sectional design made it difficult to determine causality. Longitudinal studies are required to confirm the causality of the relationship between shift work and reflux esophagitis. Second, although we used endoscopic findings and verified the classification system for the definition of reflux esophagitis, there may have been inter-operator variability in endoscopic findings. Third, the effect of shift work may vary with occupational factors such as work hours, work intensity, and work environment; these were not adequately considered here. Further research needs to consider more specific and concrete occupational factors to analyze the effects of shift work. Fourth, since our study was designed retrospectively using the results of past health examinations, we could not further analyze the dietary habits, consumption of food supplements containing melatonin, or intake of medication related to reflux esophagitis such as antibiotics, non-steroidal anti-inflammatory drugs, warfarin, and anti-hypertensives [35,49]. Fifth, our study was conducted on relatively young and well-educated Korean men and women, who were in formal full-time employment. Therefore, despite the large sample size, the generalizability of our findings to other populations needs to be verified.

Nonetheless, we believe that our study is valuable, as to the best of our knowledge, this is the first large-scale study to evaluate the association between various shift work schedules and the severity-stratified reflux esophagitis using objective endoscopic findings, as opposed to relying on self-reported symptoms. In addition, our study effectively controlled for potential confounding factors by adjusting for multiple demographic and clinical factors.

## 5. Conclusions

This cross-sectional study verified the association between shift work (especially rotating or irregular shifts) and mild reflux esophagitis by using EGD results from a large amount of health examination data. In this study, the use of an objective measure may have supported the causality; however, further longitudinal studies are required to confirm the risk of reflux esophagitis due to shift work schedules.

## Figures and Tables

**Figure 1 ijerph-18-06189-f001:**
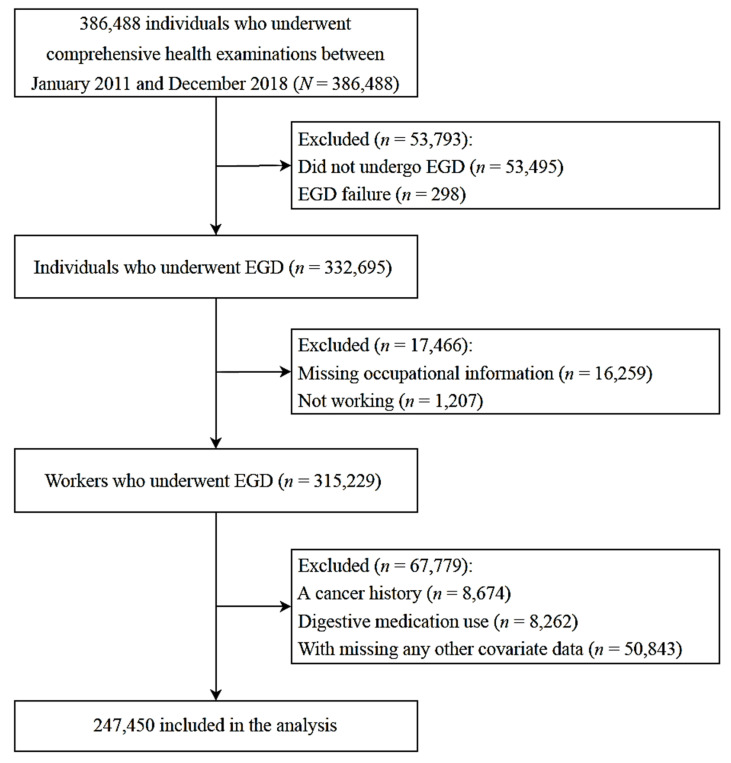
Participant selection flowchart. EGD: esophagogastroduodenoscopy.

**Table 1 ijerph-18-06189-t001:** General characteristics of the study subjects.

	Reflux Esohpagitis			
Variable Category	Total	(−)	(+)	*p*-Value
*N* (%)	247,450 (100)	197,683 (79.9)	49,767 (20.1)	
Age, years	36 (31–42)	36 (31–42)	35 (30–41)	<0.001
Sex, % male	65.0	61.9	77.5	<0.001
Current smoker, % yes	23.7	21.9	30.6	<0.001
Alcohol intake, g/day	7 (3–20)	7 (3–20)	10 (3–26)	<0.001
Exercise, % yes	39.9	39.3	42.0	<0.001
Glucose, mg/dL	93 (88–99)	93 (88–98)	94 (89–100)	<0.001
BMI, kg/m^2^	23.4 (21.2–25.6)	23.2 (21.0–25.4)	24.1 (21.8–26.4)	<0.001
Waist circumference, cm	82 (75–89)	82 (74–88)	85 (78–91)	<0.001
Total cholesterol, mg/dL	190 (169–214)	190 (169–213)	192 (171–216)	<0.001
HDL-C, mg/dL	57 (47–68)	57 (48–69)	55 (46–66)	<0.001
LDL-C, mg/dL	120 (99–143)	119 (99–142)	123 (101–146)	<0.001
Triglycerides, mg/dL	91 (64–137)	89 (63–132)	103 (71–156)	<0.001
Marital status, % married	74.1	74.4	72.8	<0.001
Education, % college or more	84.6	84.4	85.6	<0.001
Shift work (%)	23,178 (9.4) *	18,223 (9.2)	4955 (10.0)	<0.001
Fixed evening shift (%)	2132 (0.9)	1749 (0.9)	383 (0.8)	0.01
Fixed night shift (%)	577 (0.2)	472 (0.2)	105 (0.2)	0.25
Regular day and night shifts (%)	7442 (3.0)	5723 (2.9)	1719 (3.5)	<0.001
24 h shifts (%)	3232 (1.3)	2534 (1.3)	698 (1.4)	0.03
Split shift (%)	1127 (0.5)	911 (0.5)	216 (0.4)	0.43
Irregular shift (%)	3788 (1.5)	2934 (1.5)	854 (1.7)	<0.001

Abbreviations: BMI, body mass index; HDL-C, high-density lipoprotein cholesterol; LDL-C, low-density lipoprotein cholesterol. Data are presented as number (%), median (interquartile range) or percentages unless otherwise indicated. * Work schedules for some shift workers are not available (*n =* 4879, 1.9%).

**Table 2 ijerph-18-06189-t002:** The distribution of endoscopic findings of reflux esophagitis.

		RE Classification				
	Total *N*	Normal	LA-M	LA-A	LA-B	≥LA-C
***N* (%)**	247,450 (100)	197,683 (79.9)	31,132 (12.6)	16,213 (6.6)	2333 (0.9)	89 (0.04)
**Sex**						
**Male, *n* (%)**	160,829 (100)	122,269 (76.0)	21,356 (13.3)	14,906 (9.3)	2210 (1.4)	88 (0.1)
**Female, *n* (%)**	86,621 (100)	75,414 (87.1)	9776 (11.3)	1307 (1.5)	123 (0.1)	1 (0.001)
**Age**						
**18–29, *n* (%)**	42,249 (100)	33,435 (79.1)	6096 (14.4)	2470 (5.8)	238 (0.6)	10 (0.02)
**30–39, *n* (%)**	12,2301 (100)	96,949 (79.3)	15,954 (13.0)	8298 (6.8)	1072 (0.9)	28 (0.02)
**40–49, *n* (%)**	59,243 (100)	47,646 (80.4)	6931 (11.7)	3946 (6.7)	684 (1.2)	36 (0.1)
**50–59, *n* (%)**	18,710 (100)	15,499 (82.8)	1766 (9.4)	1184 (6.3)	248 (1.3)	13 (0.1)
**≥60, *n* (%)**	4947 (100)	4154 (84.0)	385 (7.8)	315 (6.4)	91 (1.8)	2 (0.04)

Abbreviations: RE, reflux esophagitis; LA, Los Angeles classification. All *p* values for trends for each RE type and normal findings were <0.001 according to sex and age.

**Table 3 ijerph-18-06189-t003:** Association between shift work and reflux esophagitis.

Work Type	Crude Model	Age and Sex Adjusted Model	Multivariate-Adjusted Models	
			Model 1	Model 2
Fixed day work	1.00 (Ref.)	1.00 (Ref.)	1.00 (Ref.)	1.00 (Ref.)
Shift work	1.09 (1.05–1.13)	1.18 (1.14–1.22)	1.15 (1.11–1.20)	1.15 (1.11–1.19)

Model 1: adjusted for age, sex, smoking status, alcohol intake, exercise, marital status, and education level. Model 2: further adjusted for metabolic factors of glucose, BMI, waist circumference, total cholesterol, HDL-C, LDL-C, and triglycerides. Abbreviations: BMI, body mass index; HDL-C, high-density lipoprotein cholesterol; LDL-C, low-density lipoprotein cholesterol; Ref., reference.

**Table 4 ijerph-18-06189-t004:** Association between shift work schedules and types of reflux esophagitis.

	RE Classification			
Work Type	Overall	LA-M	LA-A	≥LA-B
Fixed day work	1.00 (Ref.)	1.00 (Ref.)	1.00 (Ref.)	1.00 (Ref.)
Shift work	1.15 (1.11–1.19)	1.17 (1.12–1.21)	1.11 (1.05–1.18)	1.14 (0.98–1.33)
Fixed evening shift	1.10 (0.98–1.23)	1.10 (0.97–1.25)	1.08 (0.88–1.33)	1.25 (0.78–2.00)
Fixed night shift	0.94 (0.76–1.17)	0.85 (0.64–1.11)	1.09 (0.78–1.53)	1.32 (0.65–2.68)
Regular day and night shifts	1.20 (1.13–1.27)	1.22 (1.14–1.31)	1.14 (1.03–1.26)	1.24 (0.97–1.58)
24 h shifts	1.17 (1.07–1.28)	1.20 (1.09–1.33)	1.12 (0.96–1.31)	1.03 (0.66–1.59)
Split shift	1.08 (0.93–1.26)	1.13 (0.95–1.34)	1.02 (0.77–1.35)	0.94 (0.44–1.98)
Irregular shift	1.17 (1.08–1.26)	1.14 (1.03–1.25)	1.26 (1.11–1.44)	1.05 (0.73–1.53)

Odds ratios (95% confidence intervals) were fully adjusted for age, sex, smoking status, alcohol intake, exercise, marital status, education, glucose, BMI, waist circumference, total cholesterol, HDL-C, LDL-C, and triglycerides. Abbreviations: RE, reflux esophagitis; BMI, body mass index; HDL-C, high-density lipoprotein cholesterol; LDL-C, low-density lipoprotein cholesterol; LA, Los Angeles classification.

## Data Availability

The data presented in this study are available upon request from the corresponding author. The data are not publicly available because of ethical concerns.

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
