# Peer review of "Association between Shift Work and Reflux Esophagitis: The Kangbuk Samsung Health Study"

_ijerph, 2021, doi:10.3390/ijerph18126189_

Round 1

Reviewer 1 Report

This cross-sectional study is aimed to investigate the association between shift work and reflux esophagitis. The strength of the research is the use of an objective measure for the primary endpoint evaluation, the esophagogastroduodenoscopy, which can more properly substantiate the cause-effect relationship between the exposure and the event, compared to subjective measures. To date, few studies have used objective markers of reflux esophagitis to determine the effect of shift work on reflux esophagitis, while most of the published study designs used reflux self-reported symptoms questionnaires.

As underlined by the authors in the discussion of the study results, the cross-sectional design has limitations in the determination of a causal relationship. However, the authors have tried to control the potential confounding factors by stratification. The adjusted strato-specific estimates confirmed the association between shift-work and mild reflux esophagitis.

Surely, the results obtained will have to be confirmed by a longitudinal investigation. Furthermore, in vivo studies on animal models can be useful to support the biological plausibility and to confirm the suggested mechanism of action. Considering the proposed mechanism of action, a potential confounding factor that can be controlled with eligibility criteria in the next studies is the consumption of food supplements containing melatonin.

The paper is well design and written, focused, and clear in each part. The statistical approach is adequate.

I have only minor concerns:

  1. Line n. 19: Insert a close bracket after the 1.9 value.
  2. You have already followed the Strengthening the Reporting of Observational Studies in Epidemiology (STROBE) guidelines for reporting observational studies and the STROBE checklist for a cross-sectional study (https://www.equator-network.org/wp-content/uploads/2015/10/STROBE_checklist_v4_cross-sectional.pdf). Please, insert the reference to this guideline in your manuscript.
  3. Please, add to the discussion section that another limitation of the study design is the impossibility of monitoring the esophagitis-related drug consumption, such as antibiotics, non-steroidal anti-inflammatory drugs, warfarin, and anti-hypertensives.

Author Response

Thank you for your insightful comments. We have revised our manuscript as suggested, and have mentioned another limitation of the study design based on your observations.

  1. Line n. 19: Insert a close bracket after the 1.9 value.

Response: We appreciate your suggestion, and have corrected the error accordingly. (page 1, line 19)

  1. You have already followed the Strengthening the Reporting of Observational Studies in Epidemiology (STROBE) guidelines for reporting observational studies and the STROBE checklist for a cross-sectional study (https://www.equator-network.org/wp-content/uploads/2015/10/STROBE_checklist_v4_cross-sectional.pdf). Please, insert the reference to this guideline in your manuscript.

Response: Thank you for your valuable suggestion. We have cited the STROBE guidelines as suggested; the ‘Study design and participants’ section of the Materials and Methods has been revised as follows: “This cross-sectional study investigated participants in the Kangbuk Samsung Health Study between January 2011 and December 2018 (n=386,488) and followed the Strengthening the Reporting of Observational Studies in Epidemiology (STROBE) guidelines and the STROBE checklist for cross-sectional studies [18].” (page 2, lines 62-65)

  1. Please, add to the discussion section that another limitation of the study design is the impossibility of monitoring the esophagitis-related drug consumption, such as antibiotics, non-steroidal anti-inflammatory drugs, warfarin, and anti-hypertensives.

Response: Thank you for your insightful suggestion. As suggested, the ‘Strengths and limitations’ section of the discussion has accordingly been revised, as follows: “Fourth, since our study was designed retrospectively using the results of past health examinations, we could not further analyze the dietary habits, consumption of food supplements containing melatonin, or intake of medication related to reflux esophagitis such as antibiotics, non-steroidal anti-inflammatory drugs, warfarin, and anti-hypertensives [48,49].” (page 9, lines 272-276)

Reviewer 2 Report

This is a well designed and written large-scale population trial on the prevalence of GERD in shift workers.

One question - in page 2 line 73  - exclusion criteria: why was a history of cancer considered and exclusion criteria?

Author Response

Thank you for your valuable comments. We have considered your comments carefully, and have revised the manuscript, as mentioned below:

  1. One question - in page 2 line 73 - exclusion criteria: why was a history of cancer considered and exclusion criteria?

Response: We appreciate your pertinent question. Although a history of cancer does not always directly influence the development of reflux esophagitis, it can cause serious psychological stress, depression, and anxiety 1),2), and may negatively affect occupational activity.3) Conversely, several studies have shown that cancer survivors tend to change unhealthy behaviors such as smoking and poor dietary habits, that are related to reflux esophagitis.4),5) Therefore, a history of cancer may cause a significant confounding effect on the causal relationship of the health effects of shift work; we therefore excluded subjects with a history of cancer to prevent confounding.

References

1) Linden W, Vodermaier A, Mackenzie R, Greig D. Anxiety and depression after cancer diagnosis: prevalence rates by cancer type, gender, and age. J Affect Disord. 2012 Dec 10;141(2-3):343-51. doi: 10.1016/j.jad.2012.03.025. Epub 2012 Jun 21. PMID: 22727334.

2) Cardoso G, Graca J, Klut C, Trancas B, Papoila A. Depression and anxiety symptoms following cancer diagnosis: a cross-sectional study. Psychol Health Med. 2016 Jul;21(5):562-70. doi: 10.1080/13548506.2015.1125006. Epub 2015 Dec 18. PMID: 26683266.

3) Park JH, Park JH, Kim SG. Effect of cancer diagnosis on patient employment status: a nationwide longitudinal study in Korea. Psychooncology. 2009 Jul;18(7):691-9. doi: 10.1002/pon.1452. PMID: 19021127.

4) Williams K, Steptoe A, Wardle J. Is a cancer diagnosis a trigger for health behaviour change? Findings from a prospective, population-based study. Br J Cancer. 2013 Jun 11;108(11):2407-12. doi: 10.1038/bjc.2013.254. Epub 2013 May 21. PMID: 23695026; PMCID: PMC3681023.

5) Pinto BM, Eakin E, Maruyama NC. Health behavior changes after a cancer diagnosis: what do we know and where do we go from here? Ann Behav Med. 2000 Winter;22(1):38-52. doi: 10.1007/BF02895166. PMID: 10892527.

Reviewer 3 Report

First, I would like to congratulate the authors for presenting the current manuscript. Nevertheless, there are minor comments.

  1. The introduction is lacking the background that indicated the lack of studies regarding the gastroesophageal reflux disease in the working environment.
  2. The figure 1 has a bad resolution which difficulties thee reading and understanding of this. 
  3. The results are clear presented, but I considered that the tables could be further explained based on the information that they provide. 
  4. the conclusions are too reduced, based on the previous statements regarding the value of the study, the conclusions should highlight the results. 

Author Response

Thank you for your insightful comments. We have revised our manuscript as suggested; the revisions made have been mentioned below.

  1. The introduction is lacking the background that indicated the lack of studies regarding the gastroesophageal reflux disease in the working environment.

Response: We appreciate your pertinent observations, and have revised the background accordingly; the revised text is as follows: “Nevertheless, studies on gastroesophageal reflux disease in the working environment are lacking.” (page 2, lines 47-48)

  1. The figure 1 has a bad resolution which difficulties thee reading and understanding of this.

Response: We appreciate your observations, and have accordingly replaced the image in figure 1 with a clearer image to improve readability. (page 3)

  1. The results are clear presented, but I considered that the tables could be further explained based on the information that they provide.

Response: Thank you for your valuable suggestion. We have revised the results to provide a more detailed description of the data in the tables; the revised text is as follows:

“When study participants were stratified by five age groups (18-29, 30-39, 40-49, 50-59, and ≥60 years),” (page 6, lines 166-168)

“The magnitude of association was highest in the age and sex adjusted model (OR: 1.18, 95% CI: 1.14–1.22) and lowest in the crude model (OR: 1.09 95% CI: 1.05–1.13).” (page 7, lines 175-177)

“in the full-adjusted model.” (page 7, line 189)

  1. the conclusions are too reduced, based on the previous statements regarding the value of the study, the conclusions should highlight the results.

Response: We appreciate your observations, and have revised the conclusions to highlight the results based on the value of the study; the revised text is as follows: “This cross-sectional study verified the association between shift work (especially rotating or irregular shifts) and mild reflux esophagitis by using EGD results from a large amount of health examination data. In this study, the use of an objective measure may have supported the causality; however, further longitudinal studies are required to confirm the risk of reflux esophagitis due to shift work schedules.” (page 10, lines 287-291)

Reviewer 4 Report

This is a large-scale population-based cross-sectional study, to investigate about the association between shift work and reflux esophagitis. The authors provide evidence -based large scale big data to demonstrate the relationship of shift works and the reflux esophagitis. The result of the study was similar to previous epidemiological studies, but more detailed classification of various type shift works was collected in this study. I agree that this is a valuable research paper, but I have still some minor concerns about this research as the following.

Shift work might disrupt circadian rhythms, and then altered dietary habit. The altered dietary habit was considered as the major risk factor of the reflux esophagitis. Please discuss more detail about the various types shift works with the circadian rhythm disruption, and further analyze more detail about this point.

Author Response

We appreciate your observations, and have revised the manuscript as suggested. The revisions made have been indicated below.

  1. This is a large-scale population-based cross-sectional study, to investigate about the association between shift work and reflux esophagitis. The authors provide evidence -based large scale big data to demonstrate the relationship of shift works and the reflux esophagitis. The result of the study was similar to previous epidemiological studies, but more detailed classification of various type shift works was collected in this study. I agree that this is a valuable research paper, but I have still some minor concerns about this research as the following.

Shift work might disrupt circadian rhythms, and then altered dietary habit. The altered dietary habit was considered as the major risk factor of the reflux esophagitis. Please discuss more detail about the various types shift works with the circadian rhythm disruption, and further analyze more detail about this point.

Response:  We appreciate your valuable suggestions. As correctly observed, dietary habits may be one of the important plausible underlying mechanisms affecting the association. We have revised the discussion to include this issue; the revised text is as follows: “In addition, the regularity of dietary habits may have contributed to the effects on reflux esophagitis, that differed by types of shift work. For shift workers, irregular working hours are the most influential factor on the timing of meals. As shift workers who work irregular hours have difficulty in maintaining regular eating patterns, fixed time shifts would have been more beneficial than rotating or irregular shifts for achieving regular dietary habits [44].” (page 9, lines 246-251)

As our study was designed retrospectively using past health examination data, we could not perform further analysis based on the dietary habits or other factors which may affect the association, such as use of food supplements containing melatonin or intake of medication related to reflux esophagitis.

We have revised the ‘Strengths and limitations’ section of the discussion to mention this limitation; the revised text is as follows: “Fourth, since our study was designed retrospectively using the results of past health examinations, we could not further analyze the dietary habits, consumption of food supplements containing melatonin, or intake of medication related to reflux esophagitis such as antibiotics, non-steroidal anti-inflammatory drugs, warfarin, and anti-hypertensives [48,49].” (page 9, lines 272-276)

Round 2

Reviewer 3 Report

The authors have addressed all the comments, but I considered that the introduction needs further background to support the objective and study. Please include more studies in this area and what has been analysed in the country or region.

Author Response

Thank you for your valuable suggestion. As suggested, we have revised the Introduction to provide a more detailed background for this study. Accordingly, we have revised the text as follows: “Xue et al. analyzed a total of 2,027 workers who completed a comprehensive health checkup at a university hospital in China, showing that rotating night shift workers had an increased risk of GERD symptoms than day workers (odds ratio [OR]: 3.66; 95% competence interval [CI]: 2.52–5.40) [16]. Li et al. also showed that shift work was an independent risk factor for GERD symptoms (OR: 1.38; 95% CI: 1.11–1.71) in an investigation of 15,283 outpatients at 10 hospitals in China [17].” (page 2, lines 54-60)

We also have revised the Discussion to address the previous studies’ observations and limitations properly. The revised text is as follows: “Li et al. verified that shift work is an independent risk factor associated with GERD symptoms through an analysis of 15,283 outpatients (OR: 1.38; 95% CI: 1.11–1.71). However, their study was limited by only analyzing outpatients rather than healthy workers [17]. Xue et al. obtained more generalizable results than previous studies by using health checkup data. They showed that shift work and GERD symptoms are cross-sectionally associated (OR: 3.66, 95% CI: 2.52–5.40); however, their study was limited by the fact that they could not correlate GERD symptoms to endoscopic findings [16].” (page 9, lines 266-273)

Reviewer 4 Report

The authors have replied my concerns, I agree that this manuscript is suitable for publication in this journal. 

Author Response

Thank you for your valuable evaluation. Please let us know if you have any further concerns.